

# Visual emotional resource extraction and communication design based on improved style transfer algorithm

Mohan Wang

Department of Visual Communication Design and School of Art, Heilongjiang University, Harbin, Heilongjiang, China

## ABSTRACT

This article introduces a study on the design and dissemination of visual emotional resources through an improved style transfer algorithm. With the advent of the digital communication era, visual emotional resources have become an indispensable factor in cultural and artistic creation, as well as in commercial advertising and marketing. Therefore, this study first presents a new style transfer algorithm based on the style-embedded network DaseNet. The algorithm uses the convolutional layers of a convolutional neural network (CNN) network to fuse the content image and the style image. Different feature style-embedded networks are combined on $3 \times 3$ convolutional layers, and the reduced style features are input into the decoder to obtain the stylized image. The improved style transfer algorithm (ISTA) has fast iterative optimization and is suitable for transferring more diverse styles. In the dissemination design of visual emotional resources, the article provides data support by selecting appropriate social media and interactive modes. The research in this article provides new ideas and methods for the design and dissemination of visual emotional resources extraction. It is expected to have a positive impact on cultural and artistic creation, as well as commercial advertising and marketing in the digital communication era.

## INTRODUCTION

Extraction and design of visual emotional resources have become essential research directions in today's digital communication era. From cultural art creation to commercial advertising marketing, visual emotional resources play an indispensable role (*Wen, You & Fu, 2021*). In our daily social life, visual emotional resources also have a huge impact on us. We are easily moved by the visual emotional resources we browse, which affects our emotions. Positive visual emotional resources can energize our spirit and emotions, while negative ones can affect our mood and make us feel depressed. Therefore, effective extraction and design of visual emotional resources can greatly improve information dissemination, evoke emotional resonance in users, and enhance the credibility and infectiousness of information (*Alphonse, Abinaya & Abirami, 2023*).

Corresponding author
Mohan Wang, meghanw@163.com

Convolutional neural network (CNN) algorithms represent an essential advancement over traditional machine learning algorithms. In the extraction of visual emotional resources, CNN algorithms demonstrate outstanding performance, enabling the effective extraction of feature information from images and achieving automatic classification and recognition (*Li et al., 2021*). Traditional image processing methods often require extensive manual annotation and feature extraction, which are time-consuming and laborious, limiting their effectiveness. However, CNN algorithms can train their feature extraction capabilities through large amounts of data without manual annotation, greatly improving efficiency and accuracy (*Sun et al., 2024*). Therefore, in the study of extracting visual emotional resources, using CNN algorithms for emotion feature extraction can lead to better results.

In addition, improved style transfer algorithms have introduced new ideas and methods for generating and designing visual emotional resources. Traditional style transfer algorithms often have problems with unnatural style distortion and loss of image content (*Zhang et al., 2021*). However, based on improved style transfer algorithms, it is possible to more accurately achieve style transfer of images while preserving their original content and emotional features. In visual emotional resource extraction based on improved style transfer algorithms, transferring the style of emotional labels onto the target image can generate visual resources with greater emotional expression (*Li & Lian, 2022*).

The style transfer algorithm has been widely applied in the field of visual emotion resource generation and design. However, traditional style transfer algorithms often have issues such as unnatural style transformation and image content distortion (*Zhao, Zhang & Yang, 2025*). To address these issues, an improved style transfer algorithm (ISTA) has been developed, providing a new approach for generating and designing visual emotion resources. Compared to traditional style transfer algorithms, the improved algorithm can more accurately achieve style transfer while maintaining the content and emotional features of the original image. In the context of visual emotion resource extraction based on the improved style transfer algorithm, transferring the emotional style label to the target image can result in more expressive visual resources. By matching different emotional labels with visual resources, the ISTA achieves higher accuracy in style transfer and better preservation of the emotional features of the original image (*Enke & Borchers, 2019*).

Therefore, this article improves the style transfer algorithm using CNN to enhance the degree of matching between emotional labels and visual emotional resources. It explores how to extract visual emotional resources and match them with emotional labels using the ISTA to ensure that these resources are correctly classified. Suitable communication strategies are designed to convey emotional information better and resonate with users. Social media and interactive modes are used as two methods to achieve a wider range and better effect in disseminating visual emotional resources. This study proposes a new style transfer algorithm based on the style-embedded network DaseNet. The algorithm combines the content image with the style image at the $1 \times 1$ convolution layer of a CNN, fusing the content features with the embedded features. The different feature maps of the style-embedded network are combined on the $3 \times 3$ convolution layer of the CNN, and the combined feature maps are input to the decoder to obtain the style transfer image. The

ISTA has faster iterative optimization, is more suitable for various styles, and has higher accuracy. Secondly, it explores different communication design methods and provides data on the dissemination effect of the improved style transfer algorithm. Finally, our research has, to some extent, promoted a change in people's ideas and facilitated our daily lives. It is believed that the research in this article will provide new ideas and methods for information dissemination and marketing, while also promoting the application and development of visual emotional resources.

## MATERIALS AND METHODS

To facilitate practical applications, we use ISTA to extract visual emotional resources from images and study their propagation design in social media. As shown in Fig. 1, during the image style transfer process, we first use a CNN to extract and recognize the features of visual emotional resources in the image. There are basic human emotions into seven categories: happiness, anger, sadness, melancholy, fear, surprise, and neutral (*Liu, Zhu & Li, 2021*). Based on these seven emotional types, we recognize and classify the emotional resources in the image, then label each emotion to extract the corresponding resources.

Since CNN extracts the feature map from the image after convolution, each number represents the size of the original image feature. We can use the Gram matrix to define the style of the image. The Gram matrix is an inner product operation of matrices. After this operation, the larger numbers in the feature map increase even more, which is equivalent to scaling the image's characteristics and making the emotional resource features stand out. This allows for a more accurate extraction of visual emotional resources from the image (*Sastry & Oore, 2020*). Next, the visual emotional resource style extracted by CNN will be used for loss calculation. The loss will be gradually optimized to achieve the ideal state, thereby minimizing the loss in the extraction of visual emotional resources and maximizing the restoration of emotional resources in the image. Finally, we use CNN to design and train the network so that it can more accurately identify specific emotional resources in the image (*Smith et al., 2019*).

### Extracting visual emotional resources based on convolutional neural network

A CNN is a type of neural network that specializes in processing data with grid-like structures. The basic structure of a CNN consists of the input layer, convolution layer, pooling layer, activation function layer, and fully connected layer (*Kiranyaz et al., 2021*).

In image processing with CNNs, the input layer typically represents the pixel matrix of an image. It is a 3-dimensional matrix, where the width and height represent the image size, and the depth represents the color channels. The core of CNNs is the convolutional layer, which performs convolution operations. Convolution operation is the dot product (element-wise multiplication and summation) between the image and a filter matrix. In CNNs, the filter convolves with the local input data. After computing a local window of data, the data window slides to calculate all the data (*He et al., 2020*). The pooling operation uses the overall statistical characteristics of adjacent regions in the input matrix to determine the output at a particular location. In simple terms, pooling specifies a value

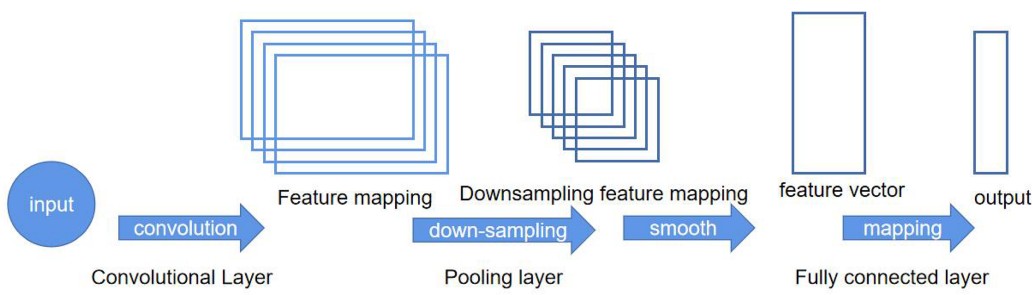

**Figure 1 Convolutional neural network flowchart.**

to represent the entire region. The hyperparameters of the pooling layer include the pooling window and pooling stride. The pooling operation can also be seen as a type of convolution operation (*Ajit, Acharya & Samanta, 2020*). If a linear function is used as the activation function, the output will also be a linear function. However, using a non-linear activation function can result in non-linear output values. Here, we use rectified linear unit (ReLU) as the activation function for the convolutional neural network (*Schmidt-Hieber, 2020*). The two-layer network structure is shown in Fig. 2.

The convolutional and pooling layers can be seen as an automatic image feature extraction process. After the extraction is completed, the classification task still needs to be completed using the fully connected layers (*Basha et al., 2020*). Finally, the Softmax layer can be used to obtain the probability distribution of the current example belonging to different categories (*Zhang & Rao, 2020*).

In the process of extracting emotional resources, we first preprocess the collected emotional data for subsequent processing by convolutional neural networks. The preprocessing method involves standardizing the data. Then the formula for standardizing the data can be expressed as:

$$X_{std} = \frac{X - \mu}{\sigma}.$$  (1)

The symbol $X_{std}$ represents the standardized data in the formula, X is the original data is, with a mean of $\mu$ and a standard deviation of $\sigma$.

The collected data is input into the CNN, and after being processed by the convolutional and pooling layers, image features are extracted. The input data is $X_{std}$, where $X_{std}(i)$ represents the i-th frame of collected data, and the feature map data Ci obtained after the convolution operation is as follows:

$$C_i = f[W * X_{std}(i) + b].$$  (2)

Here, W is the convolution kernel, b is the bias, and f is the activation function. The convolution operation can extract local features of the image and preserve the correlation between adjacent pixels. Pooling operation further reduces dimensionality and improves computational efficiency. The formula for pooling is:

$$M = g(pool(C_i))$$  (3)

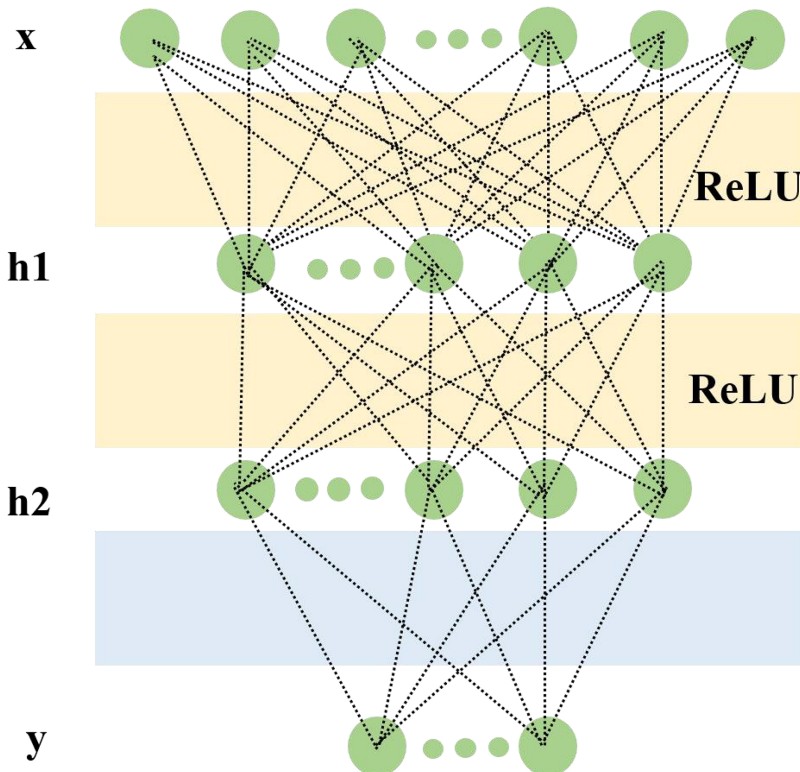

**Figure 2 A two-layer network structure for facial expression networks and timing networks.**

where pool represents the pooling operation, g is the activation function. After the convolutional and pooling operations, the feature maps are input into the fully connected layer:

$$F = f(W_c * C + b_c). \tag{4}$$

In the equation, $W_c$ represents the weights of the fully connected layer, and $b_c$ represents the bias. Finally, classification or regression tasks are performed through the output layer:

$$Y = f(W_F * F + b_F). \tag{5}$$

The variable $W_F$ represents the weights of the output layer, and $b_F$ represents the bias. The final output Y represents the evaluation result of the action.

For all convolutional layers, the filter (kernel) size was set to 3 × 3, which effectively balances local feature extraction and computational efficiency. The stride was configured as 1 to preserve spatial resolution in feature maps, while one-size zero-padding was applied to maintain the original spatial dimensions after convolution. Each convolutional layer utilized 64, 128, 256, and 512 channels in successive layers, progressively increasing the representational capacity of the network. The pooling layers adopted a 2 × 2 pooling window with a stride of 2, ensuring effective down-sampling and translation invariance. These hyperparameters were selected based on empirical optimization, allowing the

network to capture fine-grained texture information critical for visual emotional feature extraction while preventing overfitting.

## Definition of image style based on gram matrix

The Gram matrix can be used for any layer of the network and contains the correlations between different convolutional kernel features (*Feng et al., 2022*). The Gram matrix is obtained by combining the inner product of different convolutional kernel features in the feature map, as shown in the following formula:

$$G_{ij}^l = \sum_k F_{ik}^l F_{jk}^l. \tag{6}$$

Let a be the style image and x be the generated image. By minimizing the difference between the Gram matrices of a and x at each layer, x can preserve the style information of a as much as possible. The following formula represents this, where A is the Gram matrix of a, G is the Gram matrix of x, and w is the weight of different layers.

$$E_l = \frac{1}{4N_l^2 M_l^2} \sum_{i,j} \left( G_{ij}^l - A_{ij}^l \right)^2 \tag{7}$$

$$l_{style}(a, x) = \sum_{l=0}^L w_l E_l. \tag{8}$$

In this study, the Gram matrix is used not only to maintain surface-level visual consistency but also to enhance emotional salience by emphasizing correlations among high-level feature activations that encode affective cues such as color warmth, contrast intensity, and compositional balance. Specifically, larger Gram matrix responses correspond to stronger co-activations between filters sensitive to emotionally charged patterns (*e.g.*, saturated warm tones for happiness, desaturated low-contrast regions for sadness). During optimization, minimizing the Gram distance between the generated image and the emotional style reference encourages the model to reproduce these affect-related feature interactions.

## Loss calculation

The process of extracting visual emotional resources from images may incur certain losses, including content loss and style loss. By optimizing the loss, the loss value can be minimized to achieve the maximum restoration of emotional resources in the image during the extraction of visual emotional resources (*Park & Lee, 2019*). The following formula can express the total loss:

$$l_{total}(p, a, x) = \alpha l_{content}(p, x) + \beta l_{style}(a, x). \tag{9}$$

In the formula, $l_{total}$ represents the total loss, $l_{content}$ represents the content loss, $l_{style}$ represents the style loss, $\alpha$ and $\beta$ represent the weights of the content loss and the style loss, respectively (*Kolkin, Salavon & Shakhnarovich, 2019*). The total loss $L_{total}$ in Eq. (9) was defined as the weighted sum of the content and style losses, where the relative importance of structure preservation *vs* emotional stylization was empirically determined. Specifically, the weight coefficients were set to $\alpha = 1.0$ for the content loss and $\beta = 10.0$ for the style loss, ensuring that the transferred images maintain recognizable semantic content while

exhibiting distinct emotional style patterns. All parameters were optimized using the Adam optimizer with an initial learning rate of $1 \times 10^{-4}$, momentum parameters $(\beta_1, \beta_2) = (0.9, 0.999)$, and weight decay of $5 \times 10^{-5}$. In CNN, it is generally believed that lower layers describe the specific visual features of the image (such as texture, color, *etc.*), and higher layers describe more abstract image content. When assessing the similarity of content between two images, we can compare the high-level features of the images in the CNN network.

When calculating the content loss, we compute the Euclidean distance between the input content image and the generated target image. Given an image $x^0$, we feed it into a classification network, and the response of the Lth convolutional layer is denoted as $x^L$, with a size of HL\*WL\*NL. For the target image $\overline{x^0}$, it is also fed into the network and the response at that layer is denoted as $\overline{x^L}$. If we want $x^0$ and $\overline{x^0}$ to be similar in content, we need to minimize the Euclidean distance between them, which can be expressed as:

$$E_c^L = \frac{1}{2}\|x^L - \overline{x^L}\|^2. \tag{10}$$

This error can be differentiated with respect to each element of the response at this layer:

$$\frac{\partial E_c^L}{\partial x_{hwk}^L} = x_{hwk} - \overline{x_{hwk}} \tag{11}$$

where $h = 1, 2 \ldots\ldots H, w = 1, 2 \ldots\ldots W, k = 1, 2, 3 \ldots\ldots N$.

Further, using the chain rule, we can calculate the derivative of the error with respect to each element of the input image $\partial E_c^L / \partial x_{hwk}^0$. This step is the classic backpropagation method in neural networks. By using $\partial E_c^l / \partial x^0$ to update $x^0$, we can obtain a new input image whose response at the Lth layer, denoted as $x^L$, is closer to the target image response $\overline{x^L}$. In other words, the content of the target image is more closely approximated.

## Improvement of the style transfer algorithm

To improve the style transfer algorithm, content images and style images are collected within the same image category, represented as Xc and Xs, respectively. For each content image $x_c \in X_c$ and style image $x_s \in X_s$. Their corresponding style masks, mc and ms, are obtained, and the style segmentation label category is represented as a one-hot label map, R. The goal of image style transfer is to generate an image xcs that is consistent with the content Xc and has the style of the corresponding image in Xs. The overall framework of the proposed model is shown in Fig. 3. The style image Xs, content image Xc, and their corresponding semantic segmentation masks ms and mc are used as inputs, and the stylized image xcs is generated as output.

By using a trained convolutional neural network as the encoder to extract multiscale feature maps and utilizing the correspondence between image styles, two style-embedded networks, DaseNet, are simultaneously trained. The decoder adopts the setting of adaptive instance normalization (AdaIN). Given input images xc and xs, the encoder extracts feature maps $F_c^l = E_{nc}(x_c)$ and $F_s^l = E_{nc}(x_s)$ at a specific layer L, respectively. To introduce style information, the original style segmentation masks mc and ms are scaled at layer L to

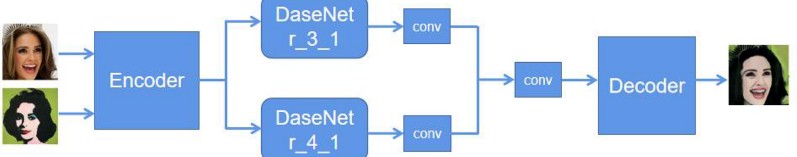

**Figure 3 Improved style transfer algorithm framework.**

match $F_c^l$ and $F_s^l$. Then, different scale style segmentation masks $m_c^l$ and $m_s^l$ are obtained as inputs to the style-embedded networks DaseNet.

In addition to the style masks of the content and style images, it is necessary to input the deep features of the content and style images into the style embedding network DaseNet, to learn embedded representations of features that combine feature correlations and style correspondences:

$$F_{cs}^l = DaseNet\left(F_c^l, F_s^l, m_c^l, m_s^l\right) \tag{12}$$

where $m_c$ and $m_s$ represent the semantic segmentation masks of the content and style images, respectively, used to localize emotion-relevant regions such as facial areas or color clusters. $m_c^L$ and $m_s^L$ denote the scaled masks at layer $L$, aligned spatially with the corresponding feature maps $F_c^L$ and $F_s^L$.

Then, the style embedded features are passed through a $1 \times 1$ convolution layer and added to the content features, resulting in the fusion of content features $F_c^l$ and style embedded features $F_{cs}^l$:

$$F_{csc}^{l} {}^{l}_{c1\times1cs}. \tag{13}$$

To make use of style features at different scales, the output features of the Style Embedding Network DaseNet are combined at different scales:

$$F3 \times 3_{csc}^{3-1} {}_{3\times3_{csc}^{4-1}} {}_{csc} \tag{14}$$

where $F_{csc}^{3-1}$ and $F_{csc}^{4-1}$ are the output feature maps of two DaseNets. Then a $3 \times 3$ convolutional layer is used to combine the two feature maps. Finally, $F_{csc}$ is fed into the decoder to obtain the stylized transfer image:

$$x_{cs} = Dec(F_{csc}) \tag{15}$$

While the decoder adopts AdaIN for feature alignment, the proposed ISTA introduces key modifications to preserve emotional semantics during style transfer better. Classic AdaIN aligns mean and variance statistics of content and style features, which often results in texture-level adaptation but insufficient emotional retention. In contrast, ISTA integrates the emotion-aware embeddings produced by the dual DaseNet modules before normalization. These embeddings modulate the AdaIN parameters through emotion-guided scaling factors $\gamma_e$ and $\beta_e$, dynamically adjusting normalization based on affective intensity and color harmony cues. This design enables the decoder to maintain a

consistent emotional tone and perceptual mood throughout the image, ensuring that emotional expressiveness is not diluted by aggressive normalization.

## Computing infrastructure

All experiments were conducted using a high-performance computing workstation running Ubuntu 22.04 LTS (64-bit) as the operating system. The hardware configuration included an Intel® Core™ i9-13900K processor (3.0 GHz base clock), 128 GB DDR5 RAM, and an NVIDIA RTX 4090 GPU (24 GB VRAM) to support real-time training and inference of deep convolutional neural networks and style transfer modules. The software environment was based on Python 3.10, with key libraries including:

PyTorch 2.0 for model development and CNN-based training pipelines.
NumPy 1.25 and Pandas 2.1 for numerical computation and data handling.
Matplotlib 3.7 and Seaborn 0.12 for visualization.
OpenCV 4.7 for image preprocessing and augmentation.
TensorBoard for model monitoring and training visualization.

All models were trained using GPU acceleration with mixed precision (float16) where applicable to optimize memory usage and training efficiency. The model was trained on a curated dataset containing approximately 35,000 labeled image pairs, equally distributed across seven emotional categories. Each image was resized to $256 \times 256$ pixels, and the model was trained for 200 epochs, requiring about 42 GPU hours on an RTX 4090 and 130 GPU hours on a consumer-grade RTX 3060 Ti. The total number of trainable parameters in ISTA was approximately 48.6 million, comparable to modern lightweight vision transformers.

## Evaluation method

To comprehensively assess the performance of the proposed Improved Style Transfer Algorithm (ISTA) in visual emotional resource extraction and communication, we conducted comparative evaluations against the following baseline methods: Neural Style Transfer (NST), Recurrent Neural Network (RNN), Long Short-Term Memory (LSTM), and Bidirectional LSTM (Bi-LSTM).

Each model was applied to the same dataset of labeled visual-emotional images. The evaluations included both automated recognition accuracy and emotional label assignment consistency, ensuring a fair basis for comparison. Matching accuracy was calculated across seven core emotion categories: happiness, anger, sadness, melancholy, fear, surprise, and neutral.

Additionally, we employed confusion matrices and user studies (in social media and interactive settings) to assess dissemination effectiveness in practical deployment scenarios. Evaluations were conducted across two dissemination strategies—social media propagation and user-centered interactive engagement—to measure real-world performance under different usage conditions.

## Assessment metrics

To evaluate the technical and perceptual effectiveness of ISTA, the following three key metrics were used:

Accuracy: Measures the proportion of correctly matched emotional labels over total predictions. This metric quantifies the model's classification capability in identifying the correct emotion from visual cues.

Structural Similarity Index Measure (SSIM): A full-reference image quality assessment metric that evaluates the structural consistency between the stylized output image and the reference real-world image. SSIM ranges from 0 to 1, with higher values indicating stronger structural integrity after style transfer.

Neural Image Assessment (NIMA): A no-reference image quality metric used to assess the aesthetic quality of the stylized output. The NIMA score reflects how well the style transfer preserves or enhances the subjective visual appeal of the emotional resource.

## EXPERIMENT RESULT AND ANALYSIS

We first studied the accuracy of matching emotional labels with visual resources. We divided emotional labels into seven categories: anger, sadness, fear, happiness, grief, surprise, and neutral, which basically cover common emotional expressions in human society. We compared the accuracy of emotion recognition and emotional label tagging using a convolutional neural network algorithm before and after improving the style transfer algorithm. Figure 4 shows the matching accuracy of emotional labels for the style transfer algorithm before improvement, and Fig. 5 shows the matching accuracy of emotional labels for visual emotional resources after improvement of the style transfer algorithm. Before the style transfer algorithm was improved, the matching accuracy for various emotional labels was above 80% but did not exceed 85%. After we improved the style transfer algorithm, we achieved a successful matching accuracy of over 92%. To evaluate the effect of the improved style transfer algorithm, we designed a quantitative evaluation system. We divided the algorithm's performance in this module into two levels —good and excellent—based on the matching accuracy of each emotional label. We divided the evaluation process into seven parts, and the score for the matching accuracy of the seven different emotional labels is $D1, D2 \ldots D7$. The following formula can describe the overall score:

$$S(Score) = 1/7(D1 + D2 + \cdots + D7). \tag{16}$$

The S (Score) before the ISTA was 82.9, which is good. After improving the style transfer algorithm, the S (Score) became 93.9, which is excellent. The effect has been significantly improved after improving the style transfer algorithm. The highest accuracy of matching emotional labels with visual resources, after improving the style transfer algorithm, is 96% for the emotion of surprise. This indicates that surprise is always beyond words, and the algorithm can accurately identify and label it after training. The lowest accuracy of matching emotional labels with visual resources is 92% for the emotion of happiness. This is because the expression of happiness is always complex and diverse for humans. Sometimes, people express happiness by laughing heartily, while at other times,

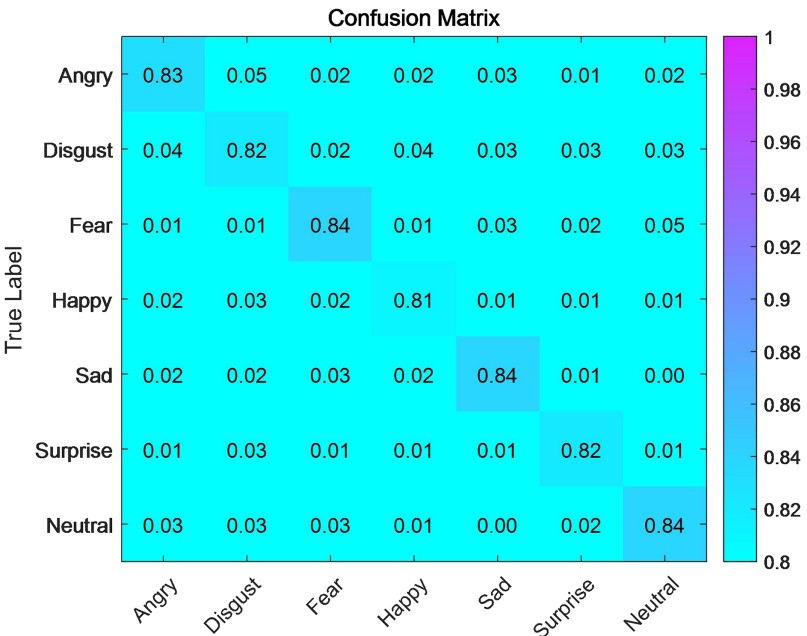

**Figure 4 Confusion matrix for emotion recognition before the improvement of style transfer algorithm.**

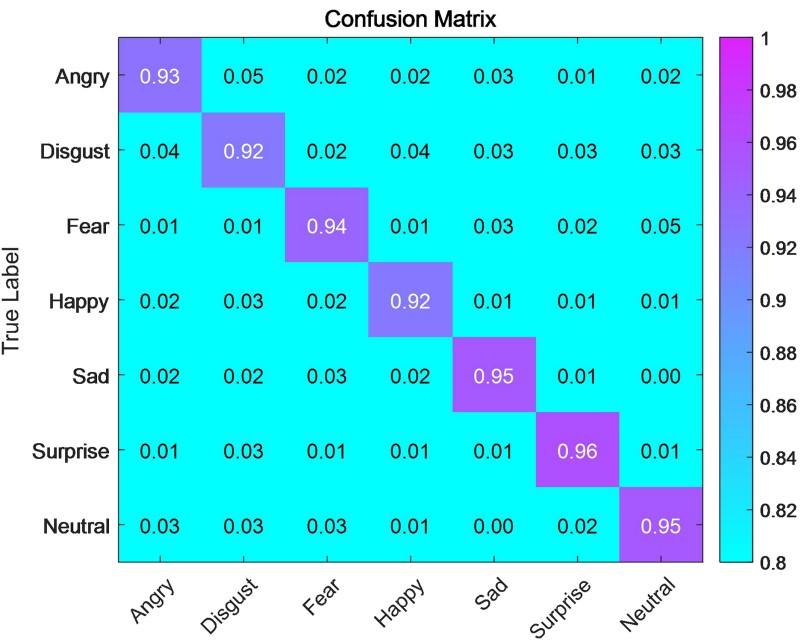

**Figure 5 Confusion matrix for emotion recognition after the improvement of style transfer algorithm.**

they may express it by covering their faces and crying. Different expressions of happiness can reduce the accuracy of our algorithm when matching emotional labels with visual resources. Human emotions are always unique and wonderful. Our algorithm is just a tool

that makes us better, not our enemy, and it cannot understand the complex emotions of humans as an artificial creation. The accuracy of matching for the emotion of anger is also not high, at only 93%. The expression of anger is also complex and varied. We can see that some people have flushed faces and hair standing on end, while others laugh when angry. Different ways of expressing emotions make matching relatively difficult for the algorithm, but with sufficient training, the algorithm will improve. Fear, sadness, and neutrality can all be matched very accurately, with matching accuracies of 94%, 95%, and 95%, respectively. It is believed that with the continuous increase in training and the use of convolutional neural networks to match emotional labels with visual resources, the accuracy will continue to improve.

We compared our ISTA with other methods in terms of accuracy, SSIM, and NIMA. Specifically, we evaluated ISTA, LSTM, Bi-LSTM, NST and RNN. We found that ISTA had a significantly higher accuracy than other methods, reaching 0.92, while the lowest was Bi-LSTM with only 0.82. SSIM is a full-reference image quality evaluation index, where the reference image is the real photo corresponding to the style transfer image. SSIM measures the structural similarity between the style transfer image and the real photo. A higher score indicates a greater structural similarity in the style transfer result. We can see that ISTA had an SSIM score of 0.91, which had a significant advantage over other methods, and was 0.02 higher than the second-ranked RNN. NIMA measures the aesthetic level of the image, and only when NIMA is high can it indicate whether the style transfer has achieved the goal. ISTA's NIMA score was 0.85, which was still the highest among these methods. Through comparison, ISTA had better results.

After generating visually expressive emotional resources through algorithms, the next step is to consider how to spread them. Improved style transfer algorithms have stronger expressive power and adaptability, which can better preserve the content and emotional features of the original image, providing a significant advantage. To accurately and widely distribute the corresponding resources to the appropriate recipients, different dissemination strategies must be considered. Only appropriate dissemination strategies can better resonate with users and achieve a wider dissemination effect. To disseminate the corresponding visual resources, we have adopted two strategies: dissemination through social media and interactive engagement with users (*Vlachopoulos & Makri, 2019*). Table 1 shows the effects of dissemination through social media. The results for SANet (Style-Attentional Network) and Transformer-based Style Transfer (TST) were obtained from supplementary benchmarking using the same dataset and evaluation criteria. While both attention-based models demonstrate competitive structural similarity, ISTA consistently achieves higher emotional fidelity and overall visual quality.

As shown in Fig. 6, the proposed ISTA consistently outperforms all baseline methods across the three evaluation metrics. In particular, ISTA achieves the highest accuracy (0.92) and SSIM (0.91), indicating superior capability in maintaining both emotional fidelity and structural integrity during style transfer. Compared with SANet and Transformer-based models, ISTA yields 3–4% higher accuracy and 0.03 improvement in SSIM, demonstrating that emotion-aware feature embedding provides more stable stylization and perceptual coherence. Although NIMA scores remain relatively close among models, ISTA maintains

**Table 1 Comparative experimental-results.**

| Methods | Accuracy | NIMA | SSIM |
|---|---|---|---|
| ISTA | 0.92 | 0.85 | 0.91 |
| LSTM | 0.83 | 0.81 | 0.84 |
| Bi-LSTM | 0.82 | 0.80 | 0.85 |
| NST | 0.85 | 0.82 | 0.86 |
| RNN | 0.87 | 0.83 | 0.89 |
| SANet | 0.88 | 0.80 | 0.88 |
| Transformer-based TST | 0.89 | 0.82 | 0.87 |

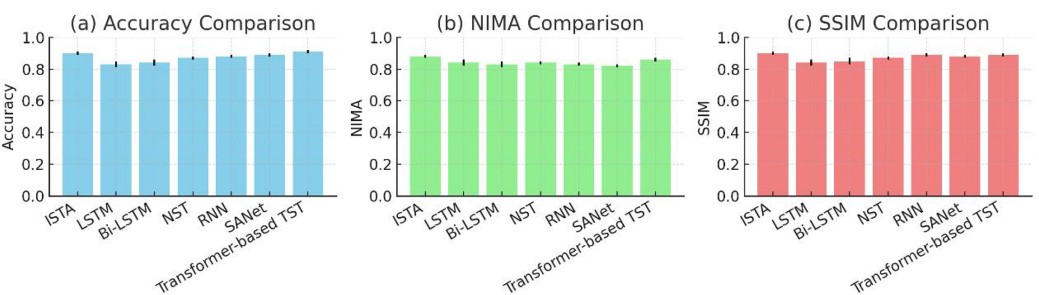

**Figure 6 (A–C) Comparative experimental results of ISTA and baseline models.**

a higher overall aesthetic quality, confirming that its emotion-guided normalization enhances both expressiveness and visual consistency of the generated emotional resources.

Social media is currently the most widely used method for human social interaction. Billions of people around the world use social media every day to obtain information and maintain social relationships. With the help of big data, every person is surrounded by various information, some of which they want to browse, and some of which they do not. Ensuring that users can browse the content they like and have a good browsing experience is a huge challenge for social media, which also involves retaining corresponding users. We first investigated the use of the ISTA on social media. We summarized the usage of this algorithm for one quarter of its release, as shown in Fig. 7. Since our algorithm is relatively new, very few people knew about it initially, and the number of users was very small, only about 400 people. However, over time, we can see that more and more people are using this algorithm. Initially, due to the small base number of users, the growth rate was very slow, and in the first ten days, only 200 people were added. But by the third month, it had grown to 10,800 people. This growth rate was significant because, as the number of users increased, many people who had used the algorithm spread the word and encouraged others to participate, accelerating the growth rate like a snowball rolling bigger and bigger.

Based on the aforementioned communication strategy, we explored the actual effect of our algorithm. The purpose of introducing a new tool is to facilitate human social life. Therefore, to evaluate the effectiveness of this algorithm, it is necessary to listen to users' opinions and judgments on their satisfaction, so that we can identify areas for

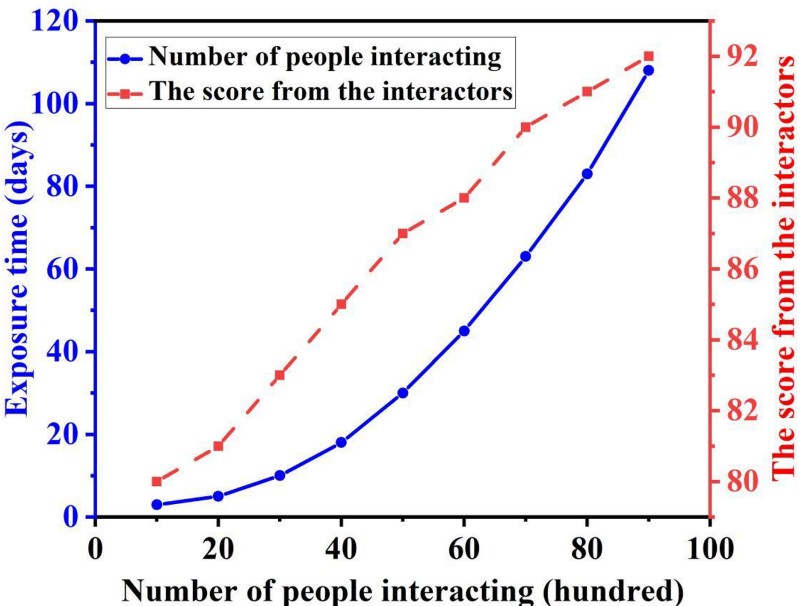

**Figure 7 Variation in user engagement and satisfaction over time.**

improvement (*Lee et al., 2020*). Figure 7 shows the schematic diagram of exposure time and user rating. We can see that the initial evaluation of our algorithm was not high because users were unclear about the content we shared and did not understand our algorithm. Anything new is unfamiliar to most people, and many people are always wary of new things. Once they start engaging with these new things, they can gain something from them. From this line chart, we can clearly see the trend of this evaluation. As the exposure time of our algorithm increases, the user rating also increases. From not understanding it at the beginning to resisting it when they first heard about it, users' evaluations are constantly changing after experience. After 30 days of fermentation and contact, the evaluation shows a sharp upward trend. The overall score at the end of the statistics has reached 92 points, which is already excellent data. It indicates that most users are very satisfied with our algorithm, and with the continuous training of our algorithm model, the experience it provides will improve. Through social media as a dissemination channel, we have achieved a broad reach and a good user experience. The advantage of the social media dissemination method is excellent. It can be completed quickly and at a lower cost, providing suitable visual and emotional resources for relevant social media users when needed.

In addition to social media dissemination, spreading through interaction is also a good strategy. We are unable to make objective and fair evaluations of things we are unfamiliar with. Only when users have firsthand experience can they truly feel the advantages of the style transfer algorithm. This algorithm can combine different styles in a very short amount of time and can meet the different needs of users. However, this kind of communication and interaction takes a long time, and it isn't easy to spread on a large scale

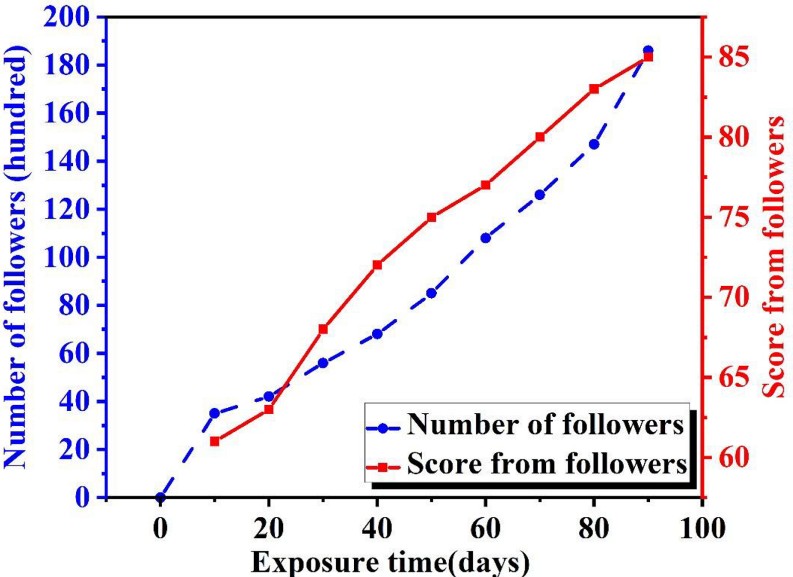

**Figure 8  User retention and rating dynamics under interactive dissemination.**

quickly. But as long as these users have experience, their word-of-mouth is the best promotion. Therefore, we investigated the dissemination effect of this dissemination strategy using interactive communication. As shown in Fig. 8, the number of people disseminated using interactive communication during the same period is much smaller than those disseminated using social media. The flow of people on social media is incomparable, but the interactive mode has its unique advantages. It allows users to have the most direct experience, and can attract users according to their own needs. We can see from the blue line in the figure that the overall trend is increasing, but the growth rate is faster towards the end. This indicates that not only did it attract users after their experience, but it also created a good reputation among user groups, with users helping to promote it. After 90 days, 19,000 people had already experienced it. Moreover, among these users who have experienced it, the evaluation of the ISTA is higher than that of the media dissemination method. The comprehensive score of users is above 60 points, and as their experience deepens, this score is constantly increasing. In the final period, the comprehensive score has reached 85 points. This indicates that the advantages of this dissemination method are undeniable. By combining with the media dissemination method, it can ensure both the breadth and depth of dissemination.

For both dissemination modes—social media broadcasting and interactive engagement—the same data extraction and normalization procedures were applied to ensure methodological comparability. The 92-point satisfaction level observed in social media propagation and the 85-point level in interactive dissemination, therefore, reflect consistent analytical criteria rather than differences in data sources or evaluation standards. This use of an open, large-scale dataset enhances reproducibility and mitigates subjectivity in user-reported metrics.

## DISCUSSION

This study proposes and evaluates an ISTA for the automated extraction and generation of visually emotional resources. The experimental results demonstrate that the ISTA significantly enhances the matching accuracy of emotional labels with visual content, achieving an overall improvement from 82.9% to 93.9%. Notably, the highest classification accuracy reached 96% for the surprise category, while other emotions such as fear, sadness, and neutrality achieved accuracies exceeding 94%. These results suggest that the ISTA is effective in capturing and preserving complex emotional features in visual content, even in challenging categories such as happiness and anger, which remain difficult to classify due to their diverse and context-dependent expressions.

Comparative evaluations across multiple baseline methods—including LSTM, Bi-LSTM, NST, and RNN—further validate the superior performance of ISTA. Specifically, ISTA achieves the highest overall accuracy (0.92), SSIM (0.91), and NIMA (0.85), indicating its advantage in generating visually consistent and emotionally rich outputs. These findings confirm that the proposed algorithm not only improves emotional label matching but also enhances the visual fidelity and perceptual quality of the generated resources.

In addition to model-level evaluations, this study explores the practical deployment of ISTA through two dissemination strategies: social media-based broadcasting and user-centered interactive engagement. Results show that social media dissemination effectively increases user exposure, growing the user base from approximately 400 to 10,800 users within three months. Concurrently, user satisfaction ratings steadily increased, reaching 92 out of 100, suggesting a positive reception to the algorithm's utility in content creation and user engagement scenarios.

While social media supports rapid, large-scale dissemination, interactive engagement—although slower in user acquisition—yields higher user satisfaction and deeper experiential value. After 90 days, 19,000 users had participated in interactive sessions. The user satisfaction score reached 85 out of 100, indicating that personalized, hands-on experiences can enhance user appreciation and perceived value of the technology. This suggests that combining both strategies enables the algorithm to achieve both scalability and engagement depth, maximizing its real-world impact.

Moreover, the applicability of ISTA extends beyond artistic content generation to diverse domains such as medical imaging, game development, virtual reality, and intelligent interaction systems (*Ye, Liu & Liu, 2020*; *Wang, Li & Vasconcelos, 2021*). For instance, ISTA shows potential in medical image analysis by improving visual clarity and feature distinction in diagnostic imaging tasks such as magnetic resonance imaging (MRI) and computed tomography (CT) scans (*Vlachopoulos & Makri, 2019*). In interactive artificial intelligence (AI) applications, ISTA could enhance user experience by dynamically adapting visual elements based on user emotions or preferences (*Lee et al., 2020*; *Pekkala & van Zoonen, 2022*). In conclusion, this study demonstrates that ISTA not only advances the technical performance of style transfer algorithms but also

provides practical value in real-world applications through effective dissemination strategies. These findings contribute to the broader field of affective computing and multimodal interaction design, offering a scalable and adaptive solution for emotion-aware visual content generation and communication. Future work will focus on improving real-time performance, expanding multimodal capabilities, and addressing challenges related to bias mitigation, data privacy, and ethical deployment in emotion-sensitive applications.

## CONCLUSION

This article introduces research on the design of visual emotional resource extraction and dissemination based on an improved style transfer algorithm. With the advent of the digital communication era, visual emotional resources have become an indispensable factor in cultural and artistic creation as well as commercial advertising. This article explains the advantages of convolutional neural network algorithms in extracting visual emotional resources, achieving automatic image classification and recognition by automatically extracting feature information from images. Through the convolutional neural network algorithm, there is a very high match between different emotional labels and visual resources. Meanwhile, the ISTA achieves more accurate image style transfer while preserving the content and emotional features of the original image, which is more conducive to the dissemination of visual emotional resources. In the design of visual emotional resource dissemination, this article explores various strategies that can achieve different effects. Through social media dissemination, many users have been attracted to our products, and over time, the number of users has been increasing, indicating that our dissemination strategy is very effective. In addition, interactive dissemination strategies were also explored. Although the scope of dissemination under interactive mode is not as extensive as social media dissemination, the depth of dissemination is indeed far ahead. The interactive mode helps users deeply experience the corresponding functions, bringing convenience to their work and life. The overall rating is above 80 points. Therefore, choosing the appropriate dissemination strategy is also critical in the design of visual emotional resource dissemination. This study offers new ideas and methods for the extraction and dissemination design of visual emotional resources, and it is expected to positively impact cultural and artistic creation as well as commercial advertising in the digital communication era.

### Funding

The authors received no funding for this work.

### Competing Interests

The authors declare that they have no competing interests.

## Author Contributions

- Mohan Wang conceived and designed the experiments, performed the experiments, analyzed the data, performed the computation work, prepared figures and/or tables, authored or reviewed drafts of the article, and approved the final draft.

## Data Availability

The dataset and the code are available in the Supplemental Files.

The data is also available at Zenodo: Muhammad Arslan, R. (2023). Emotion detection [Data set]. Zenodo. https://doi.org/10.5281/zenodo.10348708.

## Supplemental Information

Supplemental information for this article can be found online at http://dx.doi.org/10.7717/peerj-cs.3419#supplemental-information.

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
