# Peer review of "Visual emotional resource extraction and communication design based on improved style transfer algorithm"

_PeerJ Computer Science, doi:10.7717/peerj-cs.3419_

## Round 0.1 · original submission · Minor Revisions

· Academic Editor

Minor Revisions

Dear Authors,

We encourage you to address the concerns and criticisms of reviewers and resubmit your paper once you have updated it accordingly.

Best wishes,

Reviewer 1 ·

Basic reporting

The manuscript proposes an Improved Style Transfer Algorithm (ISTA) based on CNN and DaseNet to enhance the extraction and dissemination of visual emotional resources. While the approach is promising, several methodological and technical aspects require clarification and refinement.
Section 2.1 describes convolution, pooling, and fully connected layers, but hyperparameters (filter size, stride, padding, number of channels) are not reported.
The standardization formula (Eq. 1) is incomplete in notation, with missing definitions for mean and standard deviation. Precise mathematical symbols should be restored and explained .

Experimental design

The definition in Eqs. (6–8) is correct but not fully connected to emotional style extraction. The paper should clarify how the Gram matrix enhances emotional salience beyond texture preservation .
Eq. (9) defines total loss as a weighted sum of content and style loss. However, the chosen weights are not specified, nor is the optimization strategy (Adam, SGD) discussed.
Eq. (10–11) provides Euclidean distance but omits layer indices and normalization factors.
Figure 3 introduces DaseNet-based dual embedding, but equations (12–15) do not define variables (e.g., mcm_cmc, msm_sms, or one-hot label map RRR).
The decoder uses AdaIN, yet the manuscript does not analyze how ISTA differs from classic AdaIN in preserving emotional features.
Section 2.5 details high-end hardware (i9-13900K, RTX 4090), but does not specify training time, dataset size, or model complexity. Without this, it is unclear whether ISTA is practical on consumer-grade hardware .
Section 2.6 compares against LSTM, Bi-LSTM, NST, and RNN, but omits recent style transfer models such as SANet or transformer-based approaches.

Validity of the findings

Table 1 should be described and discussed in the corresponding section of the manuscript. The results presented need to be clearly explained, highlighting the relative performance of the compared techniques. Additionally, the best-performing technique should be emphasized (e.g., by using boldface) to allow readers to quickly identify the most effective approach.
The results should also be presented in graphical form (e.g., bar charts, line plots, or comparative graphs) in addition to tables. This will make performance trends easier to interpret, highlight differences between techniques more clearly, and improve the overall readability of the results section.
Figure 3 should be refined to improve clarity and readability. The variables, symbols, and components used in the figure must be clearly labeled and defined, and the overall layout should be adjusted to avoid visual clutter. Enhancing the resolution and ensuring consistency in formatting and color schemes would also make the figure more informative and easier to interpret.

·

Basic reporting

1. The experimental section provides accuracy improvements and explores social media vs. interactive dissemination strategies. While the results are compelling, several aspects of evaluation and interpretation require further development.
2. Figures 4–5 show an increase from 82.9% to 93.9%, but confusion matrices are only described narratively.
3. Table 1 compares ISTA to LSTM, Bi-LSTM, NST, and RNN. However, error bars and variance across multiple runs are not reported, making it unclear whether improvements (e.g., 0.92 vs. 0.87 accuracy) are statistically significant .
4. Figure 6 tracks user growth to 10,800 in three months, but lacks normalization for exposure level. Were increases driven by algorithm quality or external promotion? A controlled study would clarify .

Experimental design

5. The reported satisfaction rise to 92 points is promising, but the methodology of collecting ratings (sample size, scale definition, survey design) is not described, which weakens credibility .
6. Figure 7 shows 19,000 users after 90 days with an 85-point satisfaction score. However, it is unclear whether the same evaluation criteria were applied across both dissemination modes, making comparison difficult .

Validity of the findings

7. The discussion highlights the complementarity of social media (breadth) and interactive methods (depth). Quantitative trade-off metrics (e.g., retention rates, session duration) would strengthen this analysis .
8. Figures 6–7 and Table 1 lack sufficient captions (units, sampling intervals). More detailed annotations are needed to allow standalone interpretation

---

## Round 0.2 · accepted · Accept

· Academic Editor

Accept

Dear Authors,

The reviewers think that your paper now seems sufficiently improved. The paper seems ready for publication.

Best wishes,

Reviewer 1 ·

Basic reporting

All changes have been completed.

Experimental design

All changes have been completed.

Validity of the findings

All changes have been completed.

·

Basic reporting

The authors revised the paper very effectively and now it is suitable for publication.

Experimental design

Acceptable

Validity of the findings

Acceptable

Additional comments

The authors revised the paper very effectively and now it is suitable for publication.